# Comparison of Saliva and Midturbinate Swabs for Detection of SARS-CoV-2

Jianyu Lai,[a,b] Jennifer German,[b] Filbert Hong,[b] S.-H. Sheldon Tai,[b] Kathleen M. McPhaul,[b] Donald K. Milton,[b] for the University of Maryland StopCOVID Research Group

aDepartment of Epidemiology and Biostatistics, University of Maryland School of Public Health, College Park, Maryland, USA

bPublic Health Aerobiology and Biomarker Laboratory, Institute for Applied Environmental Health, University of Maryland School of Public Health, College Park, Maryland, USA

**ABSTRACT** Saliva is an attractive sample for detecting SARS-CoV-2. However, contradictory reports exist concerning the sensitivity of saliva versus nasal swabs. We followed close contacts of COVID-19 cases for up to 14 days from the last exposure and collected self-reported symptoms, midturbinate swabs (MTS), and saliva every 2 or 3 days. Ct values, viral load, and frequency of viral detection by MTS and saliva were compared. Fifty-eight contacts provided 200 saliva-MTS pairs, and 14 contacts (13 with symptoms) had one or more positive samples. Saliva and MTS had similar rates of viral detection ($P = 0.78$) and substantial agreement ($\kappa = 0.83$). However, sensitivity varied significantly with time since symptom onset. Early on (days −3 to 2), saliva had 12 times (95% CI: 1.2, 130) greater likelihood of viral detection and 3.2 times (95% CI: 2.8, 3.8) higher RNA copy numbers compared to MTS. After day 2 of symptoms, there was a nonsignificant trend toward greater sensitivity using MTS. Saliva and MTS demonstrated high agreement making saliva a suitable alternative to MTS for SARS-CoV-2 detection. Saliva was more sensitive early in the infection when the transmission was most likely to occur, suggesting that it may be a superior and cost-effective screening tool for COVID-19.

**IMPORTANCE** The findings of this manuscript are increasingly important with new variants that appear to have shorter incubation periods emerging, which may be more prone to detection in saliva before detection in nasal swabs. Therefore, there is an urgent need to provide the science to support the use of a detection method that is highly sensitive and widely acceptable to the public to improve screening rates and early detection. The manuscript presents the first evidence that saliva-based RT-PCR is more sensitive than MTS-based RT-PCR in detecting SARS-CoV-2 during the presymptomatic period – the critical period for unwitting onward transmission. Considering other advantages of saliva samples, including the lower cost, greater acceptability within the general population, and less risk to health care workers, our findings further supported the use of saliva to identify presymptomatic infection and prevent transmission of the virus.

**KEYWORDS** COVID-19, SARS-CoV-2, midturbinate swab, polymerase chain reaction, saliva

The U.S. Centers for Disease Control and Prevention (CDC) recommends the use of upper respiratory specimens, including but not limited to nasopharyngeal, midturbinate nasal, anterior nasal, and saliva specimens for the initial diagnosis of coronavirus disease 2019 (COVID-19) (1). Although nasopharyngeal swabs (NPS) are the standard for the detection of SARS-CoV-2 by most researchers, collection requires the use of trained professionals, can cause discomfort to the patients, and may pose greater risks to health care workers during sample collection (1–4). Midturbinate swabs (MTS) are sometimes used as an alternative to NPS to reduce patient discomfort and occupational exposures to health care workers (4–6). NPS is inserted into the nostril with a distance approximating that from the patient's nostril to the ear, and anterior nasal swabs are inserted to a depth of one to one and a half

Address correspondence to Filbert Hong, dmilton@umd.edu, or Donald K. Milton, dmilton@umd.edu.

The authors declare no conflict of interest.

centimeters, a MTS represents a less invasive alternative to the NPS and is inserted to a depth of about two centimeters (1). The CDC removed its preference for NPS in April of 2020 (7) and, presumably, MTS and anterior nasal swabs are utilized more widely, but this has not been studied to our knowledge. Compared to the swab-based collection, saliva is less invasive, more affordable, and can be self-collected with minimal or no supervision (1, 8, 9).

Existing studies focusing on the sensitivity of NPS compared to MTS, and NPS compared to saliva have produced contradictory results (2, 4, 10–12). Few studies directly compare saliva and MTS specimens. Furthermore, there is strong evidence that presymptomatic transmission results in higher secondary attack rates than both symptomatic and asymptomatic transmission (13, 14). However, most of the existing studies only looked at the detection of symptomatic cases after symptom onset (2, 4, 10, 11) and few looked at detection sensitivity starting with the presymptomatic period. Therefore, research that conducts a direct comparison of MTS and saliva, including an assessment of sensitivity over time (starting during the presymptomatic period), is critical to identifying optimally sensitive methods for early detection and effective control of SARS-CoV-2 transmission.

The purpose of this study was to compare the sensitivity of MTS and saliva specimens for detecting SARS-CoV-2 by actively following close contacts of COVID-19 cases and collecting MTS and saliva samples for real-time reverse transcription PCR (RT-PCR) during their postexposure quarantine period.

## RESULTS

We enrolled 58 individuals with known close contact with an active COVID-19 case. Contacts provided a total of 200 saliva and MTS pairs. The number of days of sample collection per participant ranged from one to seven. Among the contacts, 14 (24%) had at least one positive sample, including 11 with both positive saliva and MTS samples throughout follow-up. One contact had only positive saliva on 3 out of 3 samples (on days −3, 0, and 1 post-symptom onset) and 2 had only positive MTS samples. One was positive on 2 of 2 swabs (on days 7 and 10) and another was positive on 1 of 5 swabs (day 21; negative on days 14, 17, 19, and 24). Most of the participants (91%) were unvaccinated at the time of their first sample collection. Two participants were infected with the alpha variant (B.1.1.7) while all the other positive participants were infected with earlier strains of SARS-CoV-2. Most of the positive participants (92.9%) were symptomatic, whereas only one (2.3%) participant from the test negative group reported symptoms. Symptomatic participants were enrolled at −3 to 14 days since symptom onset and gave samples for up to 24 days from onset of symptoms. Symptoms were mild across the follow-up period. One participant had an oral temperature ≥ 38°C at the time of sampling, three had temperatures ≥ 37.8°C, six had temperatures ≥ 37.5°C, and all of these were in the positive group. No other significant differences were identified between the positive and negative groups (Table 1).

**Viral RNA detection in and agreement between saliva and MTS.** Among 200 pairs of saliva and MTS samples, we detected viral RNA in the saliva of 32 (16%) and the MTS of 29 (14.5%) samples. The frequency of detection was similar for both sample types ($P = 0.781$) (Table S1). Cohen's kappa demonstrated substantial agreement ($\kappa = 0.83$) with 26 (14%) positive and 165 (82.5%) negative sample pairs (Table 2). The 14 participants who became positive by either sample type during the follow-up period provided 41 saliva-MTS sample pairs, among which 71% of MTS and 78% of saliva samples were positive (Table S2), without respect to time since symptom onset. When focusing on positive participants, however, the agreement was weak ($\kappa = 0.43$ for all and $\kappa = 0.42$ for those who were symptomatic) (Table S3A and B).

**Comparison of Ct (cycle threshold) values between saliva and MTS.** Each RT-PCR contained 10 $\mu$L of heat-treated saliva sample or RNA extracted from MTS. Assuming no loss in the process, each reaction represented 7.78 $\mu$L of a saliva sample or 40 $\mu$L of the MTS eluate. The Ct values for paired samples were highly correlated (rho = 0.84, $r^2$ = 0.74, Fig. 1A). The Ct values for saliva were on average slightly but significantly greater than for MTS samples (mean difference = 0.64, $P = 0.01$) among all 58 participants (see Fig. 1B and Fig. S1), partially reflecting the difference in their input amounts.

**TABLE 1** Characteristics of the study population

| Characteristic | Never positive | Positive for MTS or saliva | All participants |
|---|---|---|---|
| No. of participants | 44 | 14 | 58 |
| No. of sample pairs | 159 | 41 | 200 |
| No. of days of sample collection per participant, median (range) | 4 (1, 7) | 3 (1, 6) | 3 (1, 7) |
| Female, N (%) | 20 (46) | 8 (57) | 28 (48) |
| Age, mean ± SD | 26.5 ± 15.5 | 27.3 ± 13.8 | 26.7 ± 15 |
| Age group, N (%) | | | |
| <18 | 2 (4) | 1 (7) | 3 (5) |
| 18-45 | 38 (86) | 11 (79) | 49 (84) |
| >45 | 4 (9) | 2 (14) | 6 (10) |
| White, N (%) | 30 (68) | 11 (79) | 41 (71) |
| BMI, mean ± SD | 25.6 ± 4.9 | 25.2 ± 4.4 | 25.5 ± 4.7 |
| Chronic respiratory illness[a], N (%) | 17 (39) | 5 (36) | 22 (38) |
| Ever smoker, N (%) | 1 (2) | 1 (7) | 2 (3) |
| Vaccination status[b], N (%) | | | |
| No vaccination | 39 (89) | 14 (100) | 53 (91) |
| ≥14 days after 1st shot | 3 (7) | 0 (0) | 3 (5) |
| ≥14 days after second shot | 2 (4) | 0 (0) | 2 (3) |
| Ever symptomatic[c], N (%) | 1 (2) | 13 (93) | 14 (24) |
| Symptomatic participants | | | |
| Days since symptom onset at enrollment, median (range) | 2 (NA[e]) | 3 (−3, 14) | 2.5 (−3, 14) |
| Overall days since symptom onset of sample collection, median (range) | 6.5 (2, 12) | 5 (−3, 24) | 5 (−3, 24) |
| Loss of taste/smell, N (%) | 0 | 2 (15) | 2 (14) |
| Median upper respiratory symptoms[d] (IQR) | 3 (1.2, 6.2) | 2 (0, 3) | 2 (0.2, 3) |
| Median lower respiratory symptoms (IQR) | 0 (0, 0) | 0 (0, 1) | 0 (0, 1) |
| Median systemic symptoms (IQR) | 0.5 (0, 1.8) | 0 (0, 1.2) | 0 (0, 1.8) |
| Median gastrointestinal symptoms (IQR) | 0 (0, 0) | 0 (0, 0) | 0 (0, 0) |
| Temperature, mean celsius ± SD | 37 ± 0.3 | 37.2 ± 0.5 | 37.2 ± 0.4 |

[a]Chronic respiratory illness = volunteers with any chronic obstructive pulmonary disease, asthma, or other lung diseases.
[b]Vaccination status was summarized at the time of participants' first sample collection. No vaccination includes 2 persons <14 days after 1st shot.
[c]Group comparison, $P < 0.05$.
[d]Symptoms at the time of each sample collection visit. Sixteen individual symptoms were rated from 0 to 3. Systemic (max score of 12) = malaise + headache + muscle/joint ache + sweats/fever/chills; gastrointestinal (max score of 12) = loss of appetite + nausea + vomit + diarrhea; lower respiratory (max score of 9) = chest tightness + shortness of breath + cough; upper respiratory (max score of 15) = runny nose + stuffy nose + sneeze + earache + sore throat.
[e]NA, not applicable.

**Relationship between days since symptom onset, probability of detection, and viral RNA copy numbers.** The Ct values among the positive symptomatic participants increased over time (days −3 through 24), along with decreasing viral RNA copy numbers. Saliva tended to have lower Ct values and higher viral RNA copy numbers compared to MTS from days −3 to 1.5, whereas MTS samples had lower Ct values and higher viral load thereafter (Fig. 2A and B).

Among symptomatic participants who had one or more positive saliva or MTS samples, the probability (sensitivity) of detecting viral RNA in saliva samples was 91% (10/11) from day −3 to day 2 (Table 3), was 89% (16/18) from day 3 through 8, and declined significantly thereafter (Fig. 2C and Fig. S2). The probability of detecting the virus in MTS samples from day −3 through day 2 was 45% (5/11), was 94% (17/18) from day 3 through 8, and then declined.

Early in the course of infection (days −3 through 2) saliva had 12 times the odds of being positive (95% CI: 1.2, 130) and 3.2 times higher viral RNA copy numbers (95% CI: 2.8, 3.8)

**TABLE 2** Viral RNA detection in paired saliva and MTS samples from all participants (N = 58)[a]

| | MTS positive | | |
|---|---|---|---|
| Saliva positive | No | Yes | Total |
| No | 165 | 3 | 168 |
| Yes | 6 | 26 | 32 |
| Total | 171 | 29 | 200 |

[a]Cohen's Kappa between the two sample types was calculated as $\kappa = 0.83$.

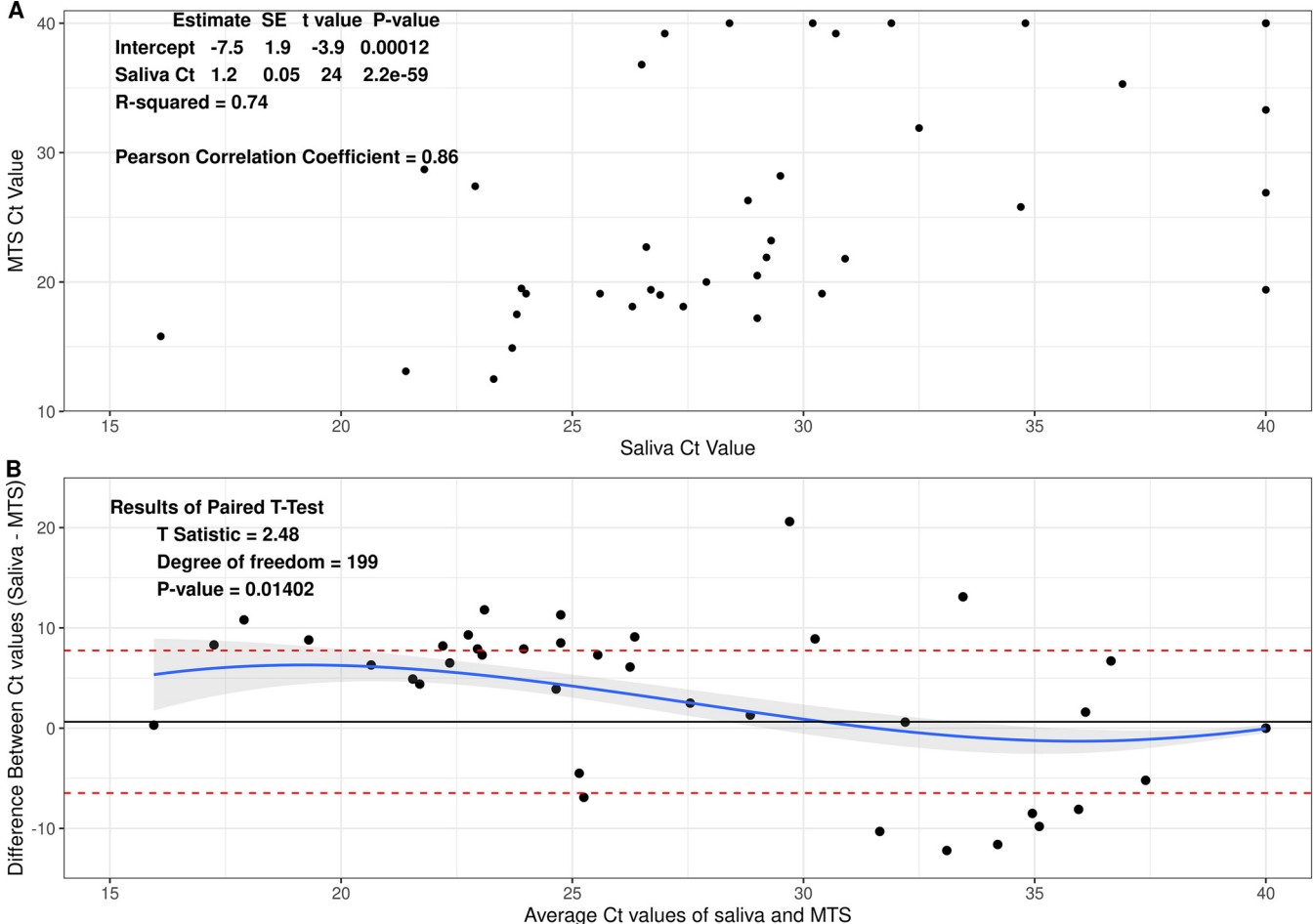

**FIG 1** Association between Ct values of saliva and MTS samples. Data were from 58 participants and 400 samples. (A) Scatterplot of Ct values of saliva and MTS. (B) Bland-Altman plot for comparison of saliva and MTS.

compared to MTS. There was a trend toward greater sensitivity and higher viral RNA copy numbers in MTS than saliva samples after day 2 post-onset of symptoms (Table 3).

**Asymptomatic case.** Only one participant from our study population was an asymptomatic case. They provided one pair of saliva and MTS samples, both of which were positive, with an average Ct value of 25.8 for MTS and 34.7 for saliva (Table S4).

## DISCUSSION

Early in the course of infection, saliva was significantly more sensitive than midturbinate nasal swabs (MTS). We found that the optimal performance of saliva was in the presymptomatic period and was more sensitive than MTS before symptom onset. Several studies have shown that presymptomatic transmission plays a more important role than symptomatic and asymptomatic transmission in the spread of SARS-CoV-2 (13, 14). Furthermore, saliva tended to have lower Ct values and higher viral load compared to MTS from the presymptomatic period through the first days after symptom onset. Together, these findings suggest that saliva may be the most effective method for detecting SARS-CoV-2 early during infection.

The CDC and the Infectious Disease Society of America recommend that COVID-19 testing allow MTS, NPS, oral swabs, anterior nasal swabs, and saliva swabs as well as saliva (1, 15). Some studies have shown differences in the sensitivity between NPS and MTS. In older, more acutely ill populations, NPS appears to be more sensitive than MTS, especially later in the course of illness (greater than 7 days) (4, 10). In a study of ambulatory and symptomatic participants whose ages were more evenly distributed, NPS and MTS swabs were highly correlated with a mean of 7 days since the onset of symptoms (16). Congrave-Wilson

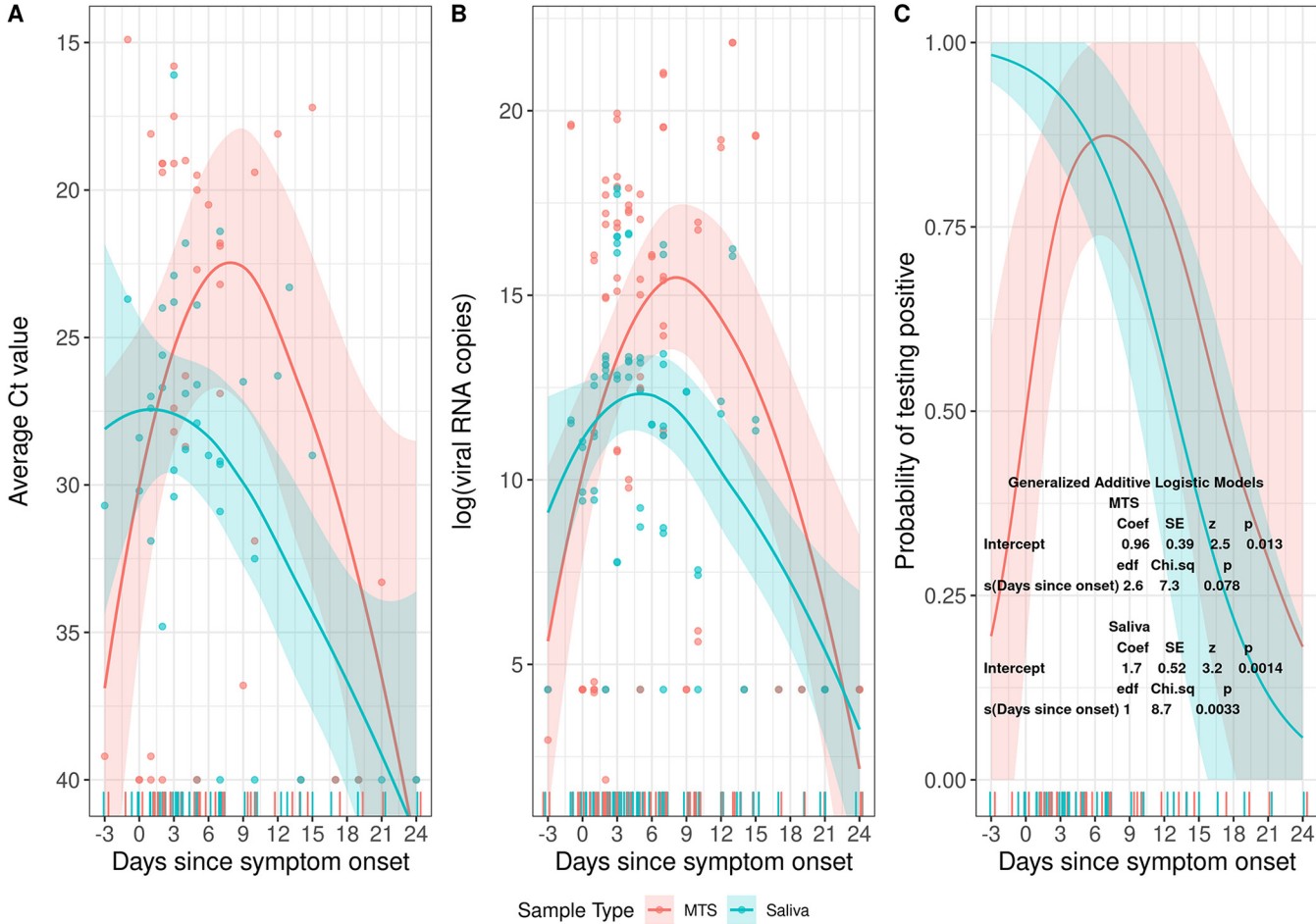

**FIG 2** The change of Ct values and probability of testing positive by days since symptom onset. Data were for MTS and saliva samples from 13 participants who provided a total of 40 pairs of samples and had one or more samples positive for SARS-CoV-2 RNA. (A) Change of Ct values by days since symptom onset. (B) Change of viral RNA copy numbers (natural log scale) by days since symptom onset. (C) Probability of being tested positive by days since symptom onset estimated from a generalized additive logistic model.

et al. (2), in agreement with the current study, found that saliva had the highest sensitivity in the first 7 days post COVID-19 onset when using NPS. Similarly, Savela et al. (17) noticed that although the peak viral load of SARS-CoV-2 in nasal swabs was higher than saliva, the latter was more likely to be positive in the first 6 days since the participants' first positive sample was detected. Becker et al. (18) compared the sensitivity of saliva and NPS for detecting COVID-19 in a convalescent cohort of 8 to 56 days since the first symptom and found that NPS performed better. They also showed that saliva was about 30% less sensitive than NPS in a separate diagnostic cohort. However, days since symptom onset were not reported,

**TABLE 3** Sensitivity of saliva and MTS and relative odds of detection and ratio of viral RNA copy numbers of SARS-CoV-2 in saliva by day since symptom onset among 13 mildly symptomatic contacts of known cases

| Days since symptom onset[a] | Saliva positive/N (Sensitivity %) | MTS positive/N (Sensitivity %) | Odds ratio[b] saliva: MTS | Estimates[c] saliva: MTS |
|---|---|---|---|---|
| All samples[d] | 31/40 (78) | 28/40 (70) | 1.5 (0.54, 4) | 0.083 (0.069, 0.099) |
| −3 through 2 | 10/11 (91) | 5/11 (45) | 12 (1.2, 130) | 3.2 (2.8, 3.8) |
| 3 through 8 | 16/18 (89) | 17/18 (94) | 0.47 (0.037, 6) | 0.03 (0.026, 0.036) |
| 9 through 24 | 5/11 (45) | 6/11 (55) | 0.7 (0.13, 3.8) | 0.065 (0.057, 0.073) |

[a]Days since symptom onset inclusive of the start and end day.
[b]Odds ratios and their 95% confidence intervals were estimated using logistic regression.
[c]Effect estimates and their 95% confidence intervals are shown as the ratio of RNA copy numbers of saliva to MTS. Analyses were controlled for random effects of subjects and sample nested within the subjects and for censoring by the limit of detection using a linear mixed-effects model for censored responses (R Project lmec package).
[d]All samples from the 13 mildly symptomatic contacts of known cases with days since symptom onset from day −3 through day 24.

so we cannot make a direct comparison with our findings. Finally, a systematic review by Bastos et al. (12) found that saliva had similar sensitivity to NPS and costs less.

Our findings have implications for improving public acceptance of COVID-19 testing, reducing the cost of mass COVID-19 screening, and improving the safety of health care workers who conduct testing. These findings are extremely important when considering large-scale screening of COVID-19 in schools and workplaces. In addition to its higher sensitivity in the early stage of the disease as demonstrated in our data, saliva has quite a few other advantages that make it an appealing screening tool. Saliva collection is less invasive and more acceptable to the general population (8, 19). One of the barriers hindering COVID-19 testing is people's fear of nasal swabs due to misinformation (20). In addition, the discomfort brought by nasal swabs may reduce people's willingness to get tested regularly, especially among children (21, 22). With the use of saliva, screening large groups with increased frequency may be more practicable. Saliva is less expensive than swab-based methods, especially if pooled samples are used (12, 23). Bastos et al. (12) estimated that using saliva saved more than $600,000 in comparison to using NPS when sampling 100,000 individuals and using a method that was more expensive than the SalivaDirect method used here. These cost savings are especially important in the context of low-resource settings.

Saliva collection is also safer for health care workers (HCWs). Amid the pandemic, one of the key concerns among HCWs is the occupational exposure to SARS-CoV-2 aerosols during some medical procedures (24). The collection of nasal swabs introduces such exposure via the close interaction between patients and HCWs and by patients' coughing and sneezing because of the procedures (25). In contrast, saliva is the only upper respiratory specimen suggested by the CDC that can be self-collected without supervision (1) and, hence, protects HCWs from directly contacting the patients when the samples are being collected. Given all these advantages of saliva compared to NPS, our findings further support the use of saliva for large-scale screening, especially of presymptomatic patients.

With that, we should also note that self-collected saliva samples may have limitations in some settings where MTS or anterior nasal swabs might be preferable. For small children or people with intellectual and developmental disabilities (IDD), the collection of saliva may require supervision or assistance (26). However, in a study with students and staff from six schools dedicated to children with IDD as the study population, investigators found that with the help of nurses and teachers, saliva samples could be collected from students for weekly testing (26). In addition, for some people, especially those who are febrile or dehydrated, the saliva samples could be thick and stringy, hence adding difficulty and additional procedures for the laboratory staff to process the samples (27, 28).

The current study has several limitations. The contacts enrolled in this study who eventually tested positive for SARS-CoV-2 developed mild, and in some cases, transient infections. Some of these mild (low viral titer) infections would not have been detected by less frequent testing protocols and may not have posed a risk for onward transmission. The sample size of those who tested positive is relatively small. Only one asymptomatic case was identified in our study so we could not compare the sensitivity of the two types of samples among asymptomatic COVID-19 cases. Previous studies demonstrated that both saliva (29) and MTS (30) were able to detect SARS-CoV-2 in asymptomatic cases in large screening programs. However, when comparing the sensitivity of saliva with that of nasal swabs, the evidence was mixed in previous analyses (2, 11) and further studies are needed to clarify this issue.

Future research should focus on the development of rapid saliva tests with high sensitivity and specificity. Tng et al. (31) proposed an amplified parallel antigen rapid test (AP-ART) using saliva to test SARS-CoV-2 with a turnaround time of only 30 min. This test was reported to have a sensitivity as high as 97%. However, the researchers did not compare this AP-ART with saliva-based RT-PCR but instead NPS-based RT-PCR and estimated the specificity of this test to be only 90%. As our study showed, this is likely an underestimate of specificity because saliva can be more sensitive than nasal swabs early in the course of the infection. Hence, further studies are needed to evaluate saliva rapid tests with a reference method that is also based on saliva.

In conclusion, the use of saliva is preferable for testing presymptomatic populations. It is more acceptable to people, which reduces barriers to testing. It is also more cost-effective for individuals to collect their saliva rather than using highly trained professionals to collect NPS and/or MTS. Finally, self-collected saliva samples eliminate the exposure to aerosols produced by sneezing, coughing, and gagging of patients undergoing NPS/MTS.

## MATERIALS AND METHODS

**Study population.** We analyzed MTS and saliva sample data from individuals who reported close contact with confirmed COVID-19 cases as part of the University of Maryland StopCOVID study (32) from May 2020 to April 2021.

**Questionnaire and sample collection.** Participants were followed every two or three days for up to 14 days from their last exposure or until SARS-CoV-2 was detected in their samples. If one or more of their screening samples became positive, results were confirmed by an appropriate clinical diagnostic test, and they were recruited to participate in the exhaled breath aspect of the study that also involved the collection of saliva and MTS (32). On each day of sample collection, participants answered an online questionnaire to update their current symptoms and medications. Those who reported having any symptoms also reported their symptom onset date (i.e., "When did you begin to feel sick?").

The symptoms checklist in the baseline and follow-up questionnaires, as previously described (32), included a runny nose, stuffy nose, sneezing, sore throat, earache, malaise, headache, muscle and/or joint ache, sweat/feverish/chills, nausea, loss of appetite, vomiting, abdominal pain or diarrhea, chest tightness, shortness of breath, and cough. Participants were self-reported for each of these 16 symptoms on a scale of 0 to 3 (0 = "no symptoms," 1 = "just noticeable," 2 = "clearly bothersome from time to time, but didn't stop me from participating in activities," 3 = "quite bothersome most or all of the time and stopped me from participating in activities").

For saliva collection, participants were instructed to not eat or drink 30 min before the visit and then collect approximately 0.5 to 1 mL of saliva drooled into a plastic collection tube. For MTS collection, trained clinical staff inserted a midturbinate swab approximately 1.5 to 2 in. into one of the participants' nostrils, rotated once, and then withdrew. This procedure was repeated in the other nostril for a total of two MTS per participant per visit.

**Laboratory analyses.** Saliva samples were processed using the SalivaDirect method (9) as previously described (32). Briefly, 50 $\mu$L of individual saliva samples were treated with proteinase K (New England Biolabs), heated at 95°C for 5 min, and kept at 4°C. MTS from both nostrils were combined and processed as previously described (32). Briefly, total nucleic acid was extracted from 200 $\mu$L of MTS with MagMax Pathogen RNA/DNA kit (Applied Biosystems) on KingFisher Duo Prime (Thermo Fisher Scientific), following the manufacturer's protocols. The sample was eluted in 50 $\mu$L of elution buffer and kept at 4°C. MS2 phage was spiked in each heat-treated saliva sample and extraction to control for extraction and PCR failure. RT-PCR was set up on the same day with each reaction consisting of 1× TaqPath 1-Step Master Mix, No ROX, 1× TaqPath COVID-19 Real-Time PCR Assay Multiplex (both from Thermo Fisher Scientific), and 10 $\mu$L of heat-treated saliva or eluted nucleic acids. Each PCR plate contained a positive-control provided in the TaqPath COVID-19 Combo kit (Thermo Fisher Scientific) and a no template control. Viral loads in saliva and MTS were quantified as previously described (32). RNA copy numbers were reported per mL for saliva and per sample for MTS. The limit of detection was 75 copies per sample and the limit of quantification was 250 copies per sample. A positive sample was defined as having Ct values <40 for at least 2 out of 3 SARS-CoV-2 targets (ORF1ab, N gene, and S gene) (33). The average Ct values of all positive targets were used in the following analyses.

**Statistical analyses.** We analyzed only paired same-day saliva and MTS samples to ensure the comparability of the two samples. Group comparisons were made between participants having a positive result for either sample and those with both samples being negative. Continuous variables (age and body mass index [BMI]) were compared using $t$ test, and categorical variables were compared using the Chi-square test (sex and chronic respiratory illness) and Fisher's exact test (age group, race, ever smoker, and vaccination status).

To compare the Ct values from saliva and MTS, we conducted paired $t$ test and Bland-Altman analysis and calculated the coefficient of determination (i.e., R squared from linear regression) and Pearson correlation coefficient. The Chi-square test was used to explore the relationship between detection and sample types. Cohen's Kappa was calculated to demonstrate the degree of agreement between the two sample types.

For participants with a positive saliva or MTS sample, we used a generalized additive logistic model (34) to estimate and plot the probability of having a positive result by days since symptom onset for the two sample types. We also created a plot using the LOESS (locally weighted smoothing) method with a 95% confidence interval for the change of Ct values and viral RNA copy numbers by days since symptom onset for the two sample types. Logistic regression was used to estimate the relative odds of detection of SARS-CoV-2 in saliva over specified intervals since symptom onset.

For the estimate of geometric means of viral RNA copy numbers and the ratio of RNA copy numbers of saliva to MTS, we applied linear mixed-effect models with censored responses (35, 36) to handle censored observations below the limit of detection and control for random effects of subjects and sample nested within subjects.

All the analyses were carried out using RStudio and R (version 4.0.4) (37).

**Ethics statement.** This study was approved by the University of Maryland Institutional Review Board and the Human Research Protection Office of the Department of the Navy. Electronically signed informed consent was obtained from all participants and questionnaire data were collected and stored with REDCap (38).

**Data availability.** The data sets generated during and/or analyzed during the current study were deposited at Open Science Framework (OSF) repository (https://osf.io/9yp3z/).

## SUPPLEMENTAL MATERIAL

Supplemental material is available online only.

**SUPPLEMENTAL FILE 1**, PDF file, 0.4 MB.

## ACKNOWLEDGMENTS

We thank all the other members of the University of Maryland StopCOVID Research Group for their efforts in recruiting participants and sample collection and processing: Oluwasanmi Oladapo Adenaiye, Barbara Albert, P. Jacob Bueno de Mesquita, Yi Esparza, Aaron Kassman, Michael Lutchenkov, Dewansh Rastogi, Maria Schanz, Isabel Sierra Maldonado, Aditya Srikakulapu, Delwin Suraj, Faith Touré, Rhonda Washington-Lewis, Somayeh Youssefi, Stuart Weston, Matthew Frieman, Mara Cai, Ashok Agrawala. We also thank Jamal Fadul and his clinic in College Park, Maryland, for assistance in recruiting study participants.

Conceptualization: D.K.M., J.G., F.H., S.-H.S.T., J.L.; data curation: F.H.; formal analysis: J.L.; investigation: S.-H.S.T., J.G., J.L., K.M.M.; methodology: D.K.M., J.G., S.-H.S.T., J.L.; project administration: D.K.M., F.H.; writing-original draft: J.L., J.G.; writing-review & editing: all authors; supervision: D.K.M.

This work was supported by Prometheus-UMD, sponsored by the Defense Advanced Research Projects Agency (DARPA) BTO under the auspices of Matthew Hepburn through agreement N66001-18-2-4015. This work was also supported by the National Institute of Allergy and Infectious Diseases Centers of Excellence for Influenza Research and Surveillance (CEIRS) Contract Number HHSN272201400008C, and the Centers for Disease Control and Prevention Contract Number 200-2020-09528. The findings and conclusions in this report are those of the authors and do not necessarily represent the official position or policy of these funding agencies and no official endorsement should be inferred.

This work was also supported by a grant from the Bill & Melinda Gates Foundation, and a generous gift from The Flu Lab (https://theflulab.org). The funders had no role in study design, data collection, analysis, decision to publish, or preparation of the manuscript.

We declare no conflict of interest.

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
