## [Reviewer comments · Microbiology Spectrum]

Microbiology Spectrum

Comparison of Saliva and Mid-Turbinate Swabs for Detection of SARS-CoV-2

Jianyu Lai, Jennifer German, Filbert Hong, Sheldon Tai, Kathleen McPhaul, and Donald Milton

Corresponding Author(s): Donald Milton, University of Maryland, College Park

Review Timeline:

Submission Date:	January 12, 2022
Editorial Decision:	February 11, 2022
Revision Received:	February 15, 2022
Accepted:	February 17, 2022

Editor: Heba Mostafa

Reviewer(s): Disclosure of reviewer identity is with reference to reviewer comments included in decision letter(s). The following individuals involved in review of your submission have agreed to reveal their identity: Delphine Dean (Reviewer #2)

Transaction Report:

DOI: <https://doi.org/10.1128/spectrum.00128-22>

February 11, 2022

Dr. Donald K Milton
University of Maryland, College Park
Maryland Institute for Applied Environmental Health
School of Public Health Building 255 SPH Rm 2302
College Park, MD 20742-2611

Re: Spectrum00128-22 (**Comparison of Saliva and Mid-Turbinate Swabs for Detection of COVID-19**)

Dear Dr. Donald K Milton:

Thank you for submitting your manuscript to Microbiology Spectrum. As you will see your paper is very close to acceptance. Please modify the manuscript along the lines I have recommended. As these revisions are quite minor, I expect that you should be able to turn in the revised paper in less than 30 days, if not sooner. If your manuscript was reviewed, you will find the reviewers' comments below.

When submitting the revised version of your paper, please provide (1) point-by-point responses to the issues I raised in your cover letter, and (2) a PDF file that indicates the changes from the original submission (by highlighting or underlining the changes) as file type "Marked Up Manuscript - For Review Only". Please use this link to submit your revised manuscript. Detailed instructions on submitting your revised paper are below.

Link Not Available

Sincerely,

Heba Mostafa

Reviewer comments:

Reviewer #1 (Comments for the Author):

The authors present a robustly designed study that provides a critical evaluation of the performance of saliva and mid-turbinate swab samples for the early detection of SARS-CoV-2. This work adds valuable data to the field and is well presented. My comments are only minor:

The authors may wish to consider 'detection of SARS-CoV-2' in instances such as the title and line 35 of the abstract (and wherever they may feel appropriate) in that they are evaluating the detection of 'SARS-CoV-2' and not just COVID-19 (the disease state).

In the introduction, the reader may also benefit from a more complete explanation of what a MTS swab is - how it compares in sampling depth as compared to both NP and AN swabs. In line with that, how frequently are MTS being collected also vs. NP and AN swabs?

Reviewer #2 (Comments for the Author):

Overall this was an interesting and very timely study looking at midturbinate nose swabs vs saliva for their suitability in diagnosing COVID-19. I think it is relevant for the journal and would have impact in the field. I have a few minor suggestions:

1. It is not clear of the participants are vaccinated/boosted or not. Patient vaccine status may impact the disease progression particularly with respect to the viral load.
2. Saliva is thought to be particularly sensitive for diagnosing certain variants (currently omicron). It would be good to indicate what variants the patients may have been infected with. While the samples themselves are perhaps not sequenced to exactly what each participant had, it would be good for discussion to talk about which variants were present in the community at the time of the study.
3. Saliva has a lot of benefit (pros). To be a bit more balanced the discussion could use some mention of some of the limitation of saliva based diagnostics. While saliva is easy to collect for most people, it can't be used for small children or patients with intellectual disabilities who cannot self collect the samples. In addition, it is more viscous and heterogeneous than swab samples in VTM so lab protocols have to be adjusted to process the saliva samples. These are not huge problems but some discussions of why MTS samples might be preferable over saliva in some settings should be included.
4. Finally, while there was only one asymptomatic patient in this study, both saliva and MTS samples have been used extensively in large scale patient screening application where they do detect and diagnose asymptomatic patients. The paper currently does not make it clear that these methods do work for asymptomatic patients in general although the sensitivity is perhaps not well studied for asymptomatic patient populations.

Preparing Revision Guidelines

- point-by-point responses to the issues I raised in your cover letter
- Upload a compare copy of the manuscript (without figures) as a "Marked-Up Manuscript" file.
- Each figure must be uploaded as a separate file, and any multipanel figures must be assembled into one file.
- Manuscript: A .DOC version of the revised manuscript
- Figures: Editable, high-resolution, individual figure files are required at revision, TIFF or EPS files are preferred

Please return the manuscript within 60 days; if you cannot complete the modification within this time period, please contact me. If you do not wish to modify the manuscript and prefer to submit it to another journal, please notify me of your decision immediately so that the manuscript may be formally withdrawn from consideration by Microbiology Spectrum.

February 17, 2022

Dr. Donald K Milton
University of Maryland, College Park
Maryland Institute for Applied Environmental Health
School of Public Health Building 255 SPH Rm 2302
College Park, MD 20742-2611

Re: Spectrum00128-22R1 (**Comparison of Saliva and Mid-Turbinate Swabs for Detection of SARS-CoV-2**)

Dear Dr. Donald K Milton:

Your manuscript has been accepted, and I am forwarding it to the ASM Journals Department for publication. You will be notified when your proofs are ready to be viewed.

Please be careful using "wild type virus "e.g. line 96". This usually refers to the original virus characterized from Wuhan early on which didn't circulate later.

Sincerely,

Heba Mostafa
Editor, Microbiology Spectrum
